# A Dual Approach to Imitation Learning from Observations with Offline Datasets

**Harshit Sikchi** *
UT Austin
hsikchi@utexas.edu

**Caleb Chuck** *
UT Austin
calebchuck@utexas.edu

**Amy Zhang**
UT Austin, Meta AI

**Scott Niekum**
UMass Amherst

**Abstract:** Demonstrations are an effective alternative to task specification for learning agents in settings where designing a reward function is difficult. However, demonstrating expert behavior in the action space of the agent becomes unwieldy when robots have complex, unintuitive morphologies. We consider the practical setting where an agent has a dataset of prior interactions with the environment and is provided with observation-only expert demonstrations. Typical *learning from observations* approaches have required either learning an inverse dynamics model or a discriminator as intermediate steps of training. Errors in these intermediate one-step models compound during downstream policy learning or deployment. We overcome these limitations by directly learning a multi-step utility function that quantifies how each action impacts the agent's divergence from the expert's visitation distribution. Using the principle of duality, we derive `DILO` (Dual Imitation Learning from Observations), an algorithm that can leverage arbitrary suboptimal data to learn imitating policies without requiring expert actions. `DILO` reduces the learning from observations problem to that of simply learning an actor and a critic, bearing similar complexity to vanilla offline RL. This allows `DILO` to gracefully scale to high dimensional observations, and demonstrate improved performance across the board. **Project page (code and videos)**: hari-sikchi.github.io/dilo/

**Keywords:** Learning from Observations, Imitation Learning

## 1 Introduction

Imitation Learning [1] promises to leverage a few expert demonstrations to train performant agents. This setting is also motivated by literature in behavioral and cognitive sciences [2, 3] that studies how humans learn by imitation, for example when mimicking other humans or watching tutorial videos. While learning from a small number of examples is often the motivation, many imitation learning methods [4, 5, 6, 7, 8] typically assume the impractical setting where the learning agent is allowed to interact with the environment as often as needed. We posit that the main reason humans can imitate efficiently is due to their knowledge priors from previous interactions with the environment; humans are able to distill skills from prior interactions to solve a desired task. Examples of expert behavior are commonly available through the ever-increasing curated, multi-robot or cross-embodied, datasets and even through tutorial videos. However, leveraging these expert datasets efficiently presents two challenges: (a) The expert data often comes in the form of observation trajectories lacking action information (e.g. tutorial videos in the observation space of agent, cross-embodiment demonstrations, etc.) (b) The learning agent should be able to leverage its collected dataset of environment interactions to mimic expert behavior. This collected dataset may not contain expert transitions and are referred to as suboptimal datasets or datasets of arbitrary quality. These challenges serve as our key motivation to bring imitation learning closer to these practical settings. We consider

---

*Equal contribution

8th Conference on Robot Learning (CoRL 2024), Munich, Germany.

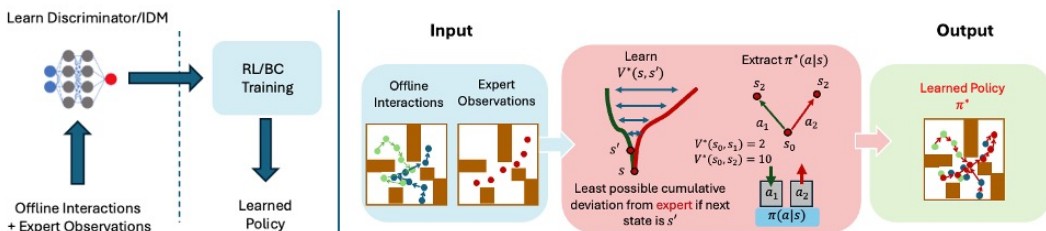

Figure 1: DILO Method Overview: Classical offline LfO methods require learning a Discriminator/IDM prior to the RL/BC step suffering from compounding errors during training/deployment respectively. DILO directly learns multi-step utility $V^*(s, s')$ of transitioning to next state in minimizing cumulative divergence with an expert avoiding errors arising due to using learned intermediate models for subsequent optimization.

the setup of offline imitation learning from observations, where the agent has access to an offline dataset of its own action-labeled transitions of arbitrary quality and is provided with potentially few task-relevant expert demonstrations in the form of observation trajectories.

Learning from Observations (LfO) has been widely studied [9, 10, 11, 12] in the online setting, where the agent is allowed to interact with the environment. A common denominator across LfO methods is the use of learned one-step models to compensate for missing expert actions. However, in the offline setting learning accurate one-step models with limited data is challenging and can result in compounding errors during downstream policy learning. These models have either taken the form of a discriminator to predict single-step expert rewards or Inverse Dynamics Models (IDM) to predict expert actions. Methods that learn a discriminator to distinguish the states or state-next states between expert and the suboptimal policy data seek to match visitation distributions of expert and the agent [8, 11, 13, 4, 14]. The learned discriminator serves as a pseudo-reward for the next step of policy optimization. In the offline setting, with limited data, the discriminator is susceptible to overfitting and any errors will compound during RL when treating the discriminator as an expert reward function [15]. A negative side-effect of using discriminator-based distribution matching in LfO is also its reliance on minimizing a proxy upper bound rather than the true objective [11, 13]. The other popular family of algorithms for LfO involves learning an IDM [16, 17], where the agent uses the offline data to predict actions from consecutive states and uses it to annotate the expert trajectories with actions. The policy is then extracted via behavior cloning on inferred expert actions. Aside from the well-known compounding error issue with behavior cloning (the errors in learned IDM only serve to exacerbate the issue), this approach discards the wealth of recovery behaviors that could be learned from offline datasets to better imitate the expert. Thus, the key question is: *Can we derive an efficient, lightweight yet principled off-policy algorithm for learning from observations that (a) learns from offline datasets of arbitrary quality, (b) bypasses the learning of intermediate one-step models, and (c) does not resort to minimizing loose upper bounds?*

In this work, we frame Imitation Learning from Observations as a modified distribution matching objective between joint state-next state visitations of the agent and expert that enables leveraging off-policy interactions. The distribution matching objective can be written as a convex program with linear constraints. Using the principle of duality, we propose Dual Imitation Learning from Observations or `DILO`, which converts the distribution matching objective to its dual form, exploiting the insight that the next state leaks information about missing actions. `DILO` no longer requires knowing expert actions in the agent's action space and instead requires sampling multiple consecutive states in the environment. An overview of our method can be found in Figure 1. `DILO` presents three key benefits over prior work: (1) `DILO` is completely off-policy and optimizes for exact distribution matching objective without resorting to minimizing upper bounds (2) `DILO` learns a multi-step utility function quantifying the effect of going to a particular next-state by minimizing long term divergence with the expert's visitation distribution, avoiding the compounding errors persistent in methods that learn intermediate one-step models. (3) `DILO` solves a single-player objective, making the learning stable and more performant. Our experimental evaluation on a suite of MuJoCo [18] environments

with offline datasets from D4RL [19] and Robomimic [20] shows that `DILO` achieves improved performance consistently over the evaluation suite. We demonstrate that `DILO` scales to image observations seamlessly without extensive hyperparameter tuning. Finally, `DILO` shows improved real robot performance compared to prior methods, which are observed to be more sensitive to the quality of the suboptimal dataset.

## 2 Related Work

**Learning from Observations**: Imitation Learning from Observations (LfO) considers the setting where the expert trajectories are available in the form of observations but missing action labels. This setting is more practical as performant algorithms developed for LfO can unlock learning from a plethora of video datasets and develop ways to transfer skills across embodiments. Unfortunately, learning from observations alone has been shown to be provably more difficult compared to the setting where expert actions are available [21]. As a result, current methods in LfO restrict themselves to small observation spaces and involve complicated learning algorithms that first train a model using offline interaction data to either predict expert actions [22, 11, 23] or learn a state-only reward function [11, 13] in the form of a discriminator. This learned model is used for subsequent Behavior Cloning, as in BCO [16], or for RL [13, 11]. As a result, prior methods suffer from compounding errors either during training or deployment. The issue of compounding errors in the offline setting with BC approaches or RL with a learned reward function has been investigated theoretically and empirically in prior works [24, 25, 15]. These errors can be fixed with repeated online interaction but can lead to substantially poor performance in the offline setting.

**Duality in RL and IL**: The duality perspective in reinforcement learning has been explored in the early works of [26, 27] and has gained recent popularity in the form of Dual RL [28, 29] and DICE [30, 31, 32, 33, 34, 35] methods. Dual approaches formulate RL as a convex program under linear constraints and leverage the Lagrangian or the Fenchel Rockefeller duality to obtain an unconstrained and principled objective for RL. The appeal of the dual perspective stems from the ability of dual approaches to learn from arbitrary off-policy data without being sensitive to distribution shift or losing sample efficiency as traditional off-policy methods [36, 37]. This behavior is attributed to the fact that dual approaches compute the on-policy policy gradient using off-policy data in contrast to traditional off-policy methods, which perform Bellman backups uniformly over state space. Duality has been previously leveraged in imitation learning from observations [35, 13, 11] by first creating an upper bound to the distribution matching objective of imitation learning such that it resembles a (return maximization) RL objective and then solving it using dual RL algorithms.

## 3 Preliminaries

We consider a learning agent in a Markov Decision Process (MDP) [38, 39] which is defined as a tuple: $\mathcal{M} = (\mathcal{S}, \mathcal{A}, p, R, \gamma, d_0)$ where $\mathcal{S}$ and $\mathcal{A}$ denote the state and action spaces respectively, $p$ denotes the transition function with $p(s'|s,a)$ indicating the probability of transitioning from $s$ to $s'$ taking action $a$; $R$ denotes the reward function and $\gamma \in (0,1)$ specifies the discount factor. The reinforcement learning objective is to obtain a policy $\pi : \mathcal{S} \to \Delta(\mathcal{A})$ that maximizes expected return: $\mathbb{E}_\pi\left[\sum_{t=0}^\infty \gamma^t r(s_t, a_t)\right]$, where we use $\mathbb{E}_\pi$ to denote the expectation under the distribution induced by $a_t \sim \pi(\cdot|s_t), s_{t+1} \sim p(\cdot|s_t, a_t)$ and $\Delta(\mathcal{A})$ denotes a probability simplex supported over $\mathcal{A}$. $f$-divergences define a measure of distance between two probability distributions given by $D_f(P\|Q) = \mathbb{E}_{x \sim Q}\left[f\left(\frac{P(x)}{Q(x)}\right)\right]$ where $f$ is a convex function.

**Visitation distributions and Dual RL**: The visitation distribution in RL is defined as the discounted probability of visiting a particular state-action under policy $\pi$, i.e $d^\pi(s,a) = (1-\gamma)\pi(a|s)\sum_{t=0}^\infty \gamma^t P(s_t = s|\pi)$ and uniquely characterizes the policy $\pi$ that achieves the visitation distribution as follows: $\pi(a|s) = \frac{d^\pi(s,a)}{\sum_a d^\pi(s,a)}$. Our proposed objective is motivated by the recently proposed Dual-V class of Dual RL [28] methods that formulates regularized RL (with conservatism

parameter $\alpha$, and offline visitation distribution $d^O$) as a convex program with state-only constraints:

$$\max_{d \geqslant 0} \ \mathbb{E}_{d(s,a)}[r(s,a)] - \alpha D_f(d(s,a) \,||\, d^O(s,a))$$

$$\text{s.t} \ \sum_{a \in \mathcal{A}} d(s,a) = (1-\gamma)d_0(s) + \gamma \sum_{(s',a') \in \mathcal{S} \times \mathcal{A}} d(s',a')p(s|s',a'), \ \forall s \in \mathcal{S}. \tag{1}$$

The above objective is constrained and difficult to optimize, but the Lagrangian dual of the above objective presents an unconstrained optimization that results in a performant Dual-RL algorithm.

$$\min_{V} (1-\gamma)\mathbb{E}_{s \sim d_0}[V(s)] + \alpha \mathbb{E}_{(s,a) \sim d^O}\left[ f_p^*\left( \left[ r(s,a) + \gamma \sum_{s'} p(s'|s,a)V(s') - V(s) \right] / \alpha \right) \right], \tag{2}$$

where $f_p^*(y) = \max_{x \in \mathbb{R}} \langle x \cdot y \rangle - f(x)$ s.t $x \geqslant 0$. Our proposed method builds upon and extend this formulation to an action-free LfO setting.

**Imitation Learning from Observations**: We consider the LfO setting where the expert provides state-only trajectories: $\mathcal{D}^{\mathcal{E}} = \{[s_0^0, s_1^0, ... s_h^0], ...[s_0^n, s_1^n, ... s_h^n]\}$. Our work focuses on the offline setting where in addition to the expert observation-trajectories, we have access to an *offline interaction data* that consists of potentially suboptimal reward-free $\{s, a, s'\}$ transitions coming from the learning agent's prior interaction with the environments. We denote the offline dataset by $d^O$ consisting of {state, action, next-state} tuples and $\rho(s, a, s')$ as the corresponding visitation distribution of the offline dataset. Distribution matching techniques aim to match the state visitation distribution of the agent to that of expert. Although we use $s$ as a placeholder for states, the method directly extends to fully-observable MDP's where we perform visitation distribution matching in the common observation space of expert and agent.

## 4 Dual Imitation Learning from Observations

Classical offline LfO approaches that rely on learning a discriminator and using it as a psuedoreward for downstream RL are susceptible to discriminator errors compounding over timesteps during value bootstrapping in RL [40, 21, 41, 15]. The discriminator is likely to overfit with limited data especially when expert observations are limited or high dimensional. Methods that learn IDM and use behavior cloning (BC) only perform policy learning on expert states and suffer compounding errors during deployment as a result of ignoring the recovery behaviors that can be extracted from offline, even suboptimal datasets [24, 25]. The key idea of the work is to propose an objective that directly learns a utility function quantifying how state transitions impact the agent's long-term divergence from the expert's visitation distribution. We derive our method below by first framing LfO as a specific visitation distribution matching problem and then leveraging duality to propose an action-free objective.

### 4.1 LfO as $\{s, s'\}$ Joint Visitation Distribution Matching

To derive our method, we first note a key observation, also leveraged by some prior works [10], that the next-state encodes the information about missing expert actions as the next-state is a stochastic function of the current state and action. We instantiate this insight in the form of a distribution matching objective. We define $\{s, s'\}$ joint visitation distributions denoted by $\tilde{d}^\pi(s, s', a') = (1 - \gamma)\pi(a'|s')\sum_{s_0 \sim d_0, a_t \sim \pi(s_t)} \gamma^t p(s_{t+1} = s', s_t = s|\pi)$. Intuitively, it extends the definition of state-action visitation distribution by denoting the discounted probability of reaching the {state, next-state} pair under policy $\pi$ and subsequently taking an action $a'$. Under this instantiation, the LfO problem reduces to finding a solution of:

$$\min_{\pi} \mathcal{D}_f(\tilde{d}^\pi(s, s', a') \| \tilde{d}^E(s, s', a')), \tag{3}$$

as at convergence, $\tilde{d}^\pi(s, s', a') = \tilde{d}^E(s, s', a')$ holds, which implies $\tilde{d}^\pi(s, s') = \tilde{d}^E(s, s')$ and also $\tilde{d}^\pi(s) = \tilde{d}^E(s)$ by marginalizing distributions. Unfortunately, the above objective (a) requires computing an on-policy visitation distribution of current policy ($\tilde{d}^\pi$) (b) provides no mechanism to incorporate offline interaction data ($d^O$), and (c) requires knowing expert actions in the action space of the agent ($a'$).

## 4.2 DILO: Leveraging Action-free Offline Interactions for Imitating Expert Observations

We now show how framing imitation (Eq. 3) as a constrained optimization objective w.r.t visitation distributions allows us to derive an action-free objective. First, in order to leverage offline interaction data $\rho$, we consider a surrogate convex mixture distribution matching objective with linear constraints:

$$\max_{\tilde{d} \geqslant 0} -\mathcal{D}_f(\mathtt{Mix}_\beta(\tilde{d}, \rho) \| \mathtt{Mix}_\beta(\tilde{d}^E, \rho))$$

$$\text{s.t} \sum_{a''} \tilde{d}(s', s'', a'') = (1-\gamma)\tilde{d}_0(s', s'') + \gamma \sum_{s,a' \in \mathcal{S} \times \mathcal{A}} \tilde{d}(s, s', a')p(s''|s', a'), \; \forall s', s'' \in \mathcal{S} \times \mathcal{S}. \quad (4)$$

The constraints above represent the *Bellman flow* conditions any valid joint visitation distribution needs to satisfy. The mixture distribution matching objective preserves the fixed point of optimization $\tilde{d}^\pi(s, s', a') = \tilde{d}^E(s, s', a')$ irrespective of mixing parameter $\beta$, thus serving as a principled objective for LfO. Mixture distribution matching has been shown to be a theoretically and practically effective way [28, 42] of leveraging off-policy data. Prior works [28, 42] dealing with state-action visitation in the context of imitation learning consider an overconstrained objective resulting in a complex min-max optimization. Our work departs by choosing constraints that are necessary and sufficient while giving us a dual objective that is *action-free* as well as a simpler *single-player optimization*. The constrained objective is convex with linear constraints. An application of Lagrangian duality to the primal objective results in the following unconstrained dual objective we refer to as DILO:

$$\mathtt{DILO:} \; \min_V \beta(1-\gamma)\mathbb{E}_{\tilde{d}_0}\big[V(s, s')\big] + \mathbb{E}_{s,s' \sim \mathtt{Mix}_\beta(\tilde{d}^E, \rho)}\big[f_p^*\big(\gamma \mathbb{E}_{s'' \sim p(\cdot|s', a')}\big[V(s', s'')\big] - V(s, s')\big)\big]$$

$$- (1-\beta)\mathbb{E}_{s,s' \sim \rho}\big[\gamma \mathbb{E}_{s'' \sim p(\cdot|s', a')}\big[V(s', s'')\big] - V(s, s')\big], \quad (5)$$

where $V$ is the Lagrange dual variable defined as $V : \mathcal{S} \times \mathcal{S} \to \mathbb{R}$ and $f_p^*$ is a variant of conjugate $f^*$ defined as $f_p^*(x) = \max(0, f'^{-1}(x))(x) - f(\max(0, f'^{-1}(x)))$. We derive DILO objective as Theorem 6.1 in Appendix 6.1 where we also see that strong duality holds and the dual objective can recover the same optimal policy with the added benefit of being action-free. Moreover, we show that the solution to the dual objective in Equation 5, $V^*(s, s')$ represents the discounted utility of transitioning to a state $s$ from $s'$ under the optimal imitating policy that minimizes the $f$-divergence with the expert visitation [43] (Appendix 6.1.2). Intuitively, this holds as the primal objective in Eq 4 can be rewritten as the reward maximization problem $\mathbb{E}_{\mathtt{Mix}_\beta(\tilde{d}, \rho)}[r(s, s', a')]$ with $r(s, s', a') = -\frac{\mathtt{Mix}_\beta(\tilde{d}, \rho)}{\mathtt{Mix}_\beta(\tilde{d}^E, \rho)}f\big(\frac{\mathtt{Mix}_\beta(\tilde{d}, \rho)}{\mathtt{Mix}_\beta(\tilde{d}^E, \rho)}\big)$. This reward function can be thought of as penalizing the policy every time it takes an action leading to a different next state-action than the expert's implied policy in agent's action space.

An empirical estimator for the DILO objective in Eq. 5 only requires sampling $s, s', s''$ under a mixture offline dataset and expert dataset and no longer requires knowing any of the actions that induced those transitions. This establishes DILO as a principled action-free alternative to optimizing the occupancy matching objective for offline settings.

## 4.3 Policy Extraction and Practical Algorithm

To instantiate our algorithm, we use the Pearson Chi-square divergence $\big(f(x) = (x-1)^2\big)$ which has been found to lead to stable DICE and Dual-RL algorithms in the past [7]. With the Pearson chi-square divergence, $f_p^*$ takes the form $f_p^*(x) = x * \big(\max\big(\frac{x}{2} + 1\big), 0\big) - \big(\big(\max\big(\frac{x}{2} + 1\big), 0\big) - 1\big)^2$. We outline the intuition of the resulting objective after substituting Pearson chi-square divergence in Appendix 6.1.4.

At convergence, the DILO objective does not directly give us the optimal policy $\pi^*$ but rather provides us with a utility function $V^*(s, s')$ that quantifies the utility of transitioning to state $s'$ from $s$ in visitation distribution matching. To recover the policy, we use value-weighted regression on the offline interaction dataset, which has been shown [44, 45, 46] to provably maximize the $V$ function (thus taking action to minimize divergence with expert's visitation) while subject to distribution constraint of offline dataset:

$$\mathcal{L}(\psi) = -\mathbb{E}_{s,a,s' \sim \rho}\Big[e^{\tau V^*(s,s')} \log \pi_\psi(a|s)\Big]. \quad (6)$$

**Choice of** $\tilde{d}_0(s, s')$: Any distribution over state and next-state is implicitly dependent on the policy that induces the next-state. The initial distribution in Eq. 5 forms the distribution over states from which the learned policy will acquire effective imitation behavior to mimic the expert. In our work, we set $\tilde{d}_0(s, s')$ to be the uniform distribution over replay buffer $\{s, s'\}$ pairs, ensuring that the learned policy is robust enough to imitate from any starting transition observed from all the transitions available to us.

**Practical optimization difficulty of dual objectives**:
Prior works in reinforcement learning that have leveraged a dual objective based on Bellman-flow constraints suffer from learning instabilities under gradient descent. Intuitively, in our case, learning instability arises as the gradients from $V(s, s')$ and $V(s', s'')$ can conflict if the network learns similar feature representations for nearby states due to feature co-adaptation [47]. Prior works [28] have resorted to using semi-gradient approaches but do not converge provably to the optimal solution [29]. To sidestep this issue, we leverage the orthogonal gradient

---
**Algorithm 1:** DILO

1: Init $V_\phi$, $\pi_\psi$
2: Params: temperature $\tau$, mixture ratio $\beta$
3: Let $\mathcal{D} = \widehat{\rho} = \{(s, a, s')\}$ be an offline dataset and $\mathcal{D}^\mathcal{E} = \{s, s'\}$ be expert demonstrations dataset.
4: **for** $t = 1..T$ iterations **do**
5:   Train $V_\phi$ via Orthogonal gradient update on Eq. 5
6:   Update $\pi_\psi$ by minimizing Eq. 6
7: **end for**

---

update proposed by ODICE [29] for the offline RL setting that fixes the conflicting gradient by combining the projection of the gradient of $V(s', s'')$ on $V(s, s')$ and the orthogonal component in a principled manner. We refer to the ODICE work for detailed exposition. Our complete practical algorithm can be found in Algorithm 1.

## 5 Experiments

In our experiments, first, we aim to understand where the prior LfO methods based on IDM or a discriminator fail and how the performance of DILO compares to baselines under a diverse set of datasets. Our experiments with proprioceptive observations consider an extensive set of 24 datasets. The environments span locomotion and manipulation tasks, containing complex tasks such as 24-DoF dextrous manipulation. Second, we examine if the simplicity of DILO objective indeed enables it to scale directly to mimic expert image observation trajectories. Finally, we test our method on a set of real-robot manipulation tasks where we consider learning from a few expert observations generated by human teleoperation as well as cross-embodied demos demonstrated by humans as videos.

### 5.1 Offline Imitation from Observation Benchmarking

We use offline imitation benchmark task from [28, 13] where the datasets are sourced from D4RL [19, 18]. For locomotion tasks, the benchmark uses an offline interaction dataset consisting of 1-million transitions from random or medium datasets mixed with 200 expert trajectories (or 30 expert trajectory in the few-expert setting). For manipulation environments, the suboptimal datasets comprise of 30 expert trajectories mixed with human or cloned datasets from D4RL. The expert demonstrates 1 observation trajectory for all tasks. DILO uses a single set of hyperparameters across all environments. Hyperparameters and additional experimental details can be found in Appendix 6.2.1.

**Baselines**: We compare DILO against offline imitation from observations (LfO) methods such as ORIL [48], SMODICE [13] as well as offline imitation from action-labelled demonstration (LfD) methods like BC [49], IQ-Learn [50] and ReCOIL [28]. We choose these imitation learning methods as they represent the frontier of the LfO and LfD setting, outperforming methods like ValueDICE [42] and DemoDICE [34] as shown in prior works. Intuitively, the imitation from action-labeled demonstrations represents the upper bound of performance as they have additional information on expert actions even though sometimes we observe LfO algorithms to surpass them in performance. ORIL and SMODICE first learn a discriminator and, subsequently run downstream RL treating the discriminator as the expert pseudo-reward.

Table 1 shows the cumulative return of different algorithms under the ground truth expert reward function that is unavailable to the learning agent during training. DILO demonstrates improved performance across a wide range of datasets. Particularly in the setting of few-expert observations or high dimensional observations like dextrous manipulation, the performance of methods relying

| Suboptimal Dataset | Env | Access to expert actions | | | | | No expert actions | | | Expert |
|---|---|---|---|---|---|---|---|---|---|---|
| | | RCE | BC expert data | BC full dataset | IQ-Learn (offline) | ReCOIL | ORIL | SMODICE | DILO | |
| random+ expert | hopper | 51.41±38.63 | 4.52±1.42 | 5.64±4.83 | 1.85±2.19 | 108.18±3.28 | 75.21±21.90 | **100.46±0.64** | 97.87±8.11 | 111.33 |
| | halfcheetah | 64.19±11.06 | 2.2±0.01 | 2.25±0.00 | 4.83±7.99 | 47.65±16.95 | 60.49±3.53 | 85.16±3.62 | **91.18±0.24** | 88.83 |
| | walker2d | 20.90±26.80 | 0.86±0.61 | 0.91±0.5 | 0.57±0.09 | 102.16±7.19 | 27.02±23.49 | **108.41±0.47** | 108.42±0.64 | 106.92 |
| | ant | 105.38±14.15 | 5.17±5.43 | 30.66±1.35 | 42.23±20.05 | 126.74±4.63 | 54.19±27.60 | **122.56±4.47** | 122.15±5.15 | 130.75 |
| random+ few-expert | hopper | 25.31±18.97 | 4.84±3.83 | 3.0±0.54 | 1.37±1.23 | 97.85±17.89 | 29.86±22.60 | 78.80±3.09 | **93.73±7.59** | 111.33 |
| | halfcheetah | 2.99±1.07 | -0.93±0.35 | 2.24±0.01 | 1.14±1.94 | 76.92±7.53 | 25.76±9.52 | 4.10±1.50 | **52.32±10.72** | 88.83 |
| | walker2d | 40.49±26.52 | 0.98±0.83 | 0.74±0.20 | 0.39±0.27 | 83.23±19.00 | 36.52±9.37 | **107.18±1.87** | 108.42±0.25 | 106.92 |
| | ant | 67.62±15.81 | 0.91±3.93 | 35.38±2.66 | 32.99±3.12 | 67.14±8.30 | -8.89±39.12 | | **117.50±4.75** | 130.75 |
| medium+ expert | hopper | 58.71±34.06 | 16.09±12.80 | 59.25±3.71 | 12.90±24.00 | 88.51±16.73 | 14.15±18.24 | 54.28±3.78 | **99.97±12.62** | 111.33 |
| | halfcheetah | 65.14±13.82 | -1.79±0.22 | 42.45±0.42 | 25.67±20.82 | 81.15±2.84 | 65.28±7.17 | 56.91±4.08 | **90.47±0.64** | 88.83 |
| | walker2d | 96.24±14.04 | 2.43±1.82 | 72.76±3.82 | 59.37±30.14 | 108.54±1.81 | 28.32±27.82 | 3.11±2.41 | **77.16±6.96** | 106.92 |
| | ant | 86.14±38.59 | 0.86±7.42 | 95.47±10.37 | 37.17±41.15 | 120.36±7.67 | 49.14±14.92 | **103.67±3.44** | 102.89±3.57 | 130.75 |
| medium few-expert | hopper | 66.15±35.16 | 7.37±1.13 | 46.87±5.31 | 11.05±20.59 | 50.01±10.36 | 11.67±14.82 | **44.61±6.08** | 41.80±14.81 | 111.33 |
| | halfcheetah | 61.14±18.31 | -1.15±0.06 | 42.21±0.06 | 26.27±20.24 | 75.96±4.54 | 59.11±4.74 | 44.66±0.95 | **74.71±6.35** | 88.83 |
| | walker2d | | 2.02±0.72 | 70.42±2.86 | 73.30±2.85 | 91.25±17.63 | 6.81±6.76 | 6.00±6.69 | **66.64±6.05** | 106.92 |
| | ant | 67.95±36.78 | -10.45±1.63 | 81.63±6.67 | 35.12±50.56 | 110.38±10.96 | 67.18±30.45 | **90.30±2.23** | 88.03±9.01 | 130.75 |
| cloned+expert | pen | 19.60±11.40 | 13.95±11.04 | 34.94±11.10 | 2.18±8.75 | 95.04±4.48 | 0.92±4.51 | 13.29±13.57 | **101.36±3.48** | 106.42 |
| | door | 0.08±0.15 | -0.22±0.05 | 0.011±0.00 | 0.07±0.02 | 102.75±4.05 | -0.32±0.01 | 1.45±2.23 | **105.60±0.28** | 103.94 |
| | hammer | 1.95±3.89 | 2.41±4.48 | 5.45±7.84 | 0.27±0.02 | 95.77±17.90 | 0.26±0.01 | 0.00±0.10 | **112.55±22.10** | 125.71 |
| human+expert | pen | 17.81±5.91 | 13.83±10.76 | 90.76±25.09 | 14.29±28.82 | 103.72±2.90 | 5.76±3.85 | -3.33±0.12 | **88.61±14.30** | 106.42 |
| | door | -0.05±0.05 | -0.03±0.05 | 103.71±1.22 | 5.6±7.29 | 104.70±0.55 | -0.32±0.01 | -0.12±0.16 | **101.51±0.99** | 103.94 |
| | hammer | 5.00±5.64 | 0.18±0.14 | 122.61±4.85 | -0.32±1.38 | 125.19±3.29 | 3.11±0.04 | 0.40±0.48 | **117.52±8.55** | 125.71 |
| partial+expert | kitchen | 6.875±9.24 | 2.5±5.0 | 45.5±1.87 | 0.0±0.0 | 60.0±5.70 | 0.00±0.00 | 35.0±4.08 | **43.0±5.30** | 75.0 |
| mixed+expert | kitchen | 1.66±2.35 | 2.2±3.8 | 42.1±1.12 | 0.0±0.0 | 52.0±1.0 | 0.00±0.00 | **48.33±4.24** | 44.0±8.30 | 75.0 |

Table 1: The normalized return obtained by different offline IL (both provided with and without expert actions) methods on the D4RL suboptimal datasets with 1 expert trajectory. The mean and std are obtained over 5 random seeds. Methods with statistically significant improvement (t-test) over second best method are highlighted.

on a learned discriminator falls sharply potentially due to overfitting of discrimination that results in compounding downstream errors. `DILO` gets rid of this intermediate step, completely reducing the problem of LfO to a similar training setup as a traditional actor-critic algorithm. BC methods, representing an upper bound to BCO [16] shows poor performance even without a learned IDM.

## 5.2 Imitating from Expert Image Observations

Learning to mimic expert in the image observation space presents a difficult problem, especially in the absence of a pretrained representations. To evaluate our algorithm in this setting, we consider the Robomimic datasets [20] which gives the flexibility of choice to use image observations or the corresponding proprioceptive states for learning. Our suboptimal datasets comprises of Multi-Human (MH), Machine Generated (MG) datasets from Robomimic without access to expert trajectories. We obtain 50 expert-observation trajectories from Proficient Human (PH) datasets. This setup is more complicated as the agent has to learn expert actions purely from OOD datasets and match expert visitations. We consider the most performant LfO baseline from the previous section SMODICE [13] along with a BCO [16] baseline as BCO has shown success in scaling up to image observations [51].

Fig 2 shows the result of these approaches on 4-different datasets using both state and image observations. SMODICE shows competitive results when learning from state-observations but does not scale up well to images likely due to the overfitting of the discriminator in a high-dimensional observation space. BCO fails consistently across both state and image experiments as learning an IDM is challenging in this contact-rich task, and any mistake by IDM can compound. `DILO` outperforms baselines and demonstrates improved performance across both state and image observations.

| | | Lift-MG | Lift-MH | Can-MG | Can-MH |
|---|---|---|---|---|---|
| State 50 Demos | BCO | 0.00 ± 0.00 | 0.00 ± 0.00 | 0.00 ± 0.00 | 0.0 0± 0.00 |
| | SMODICE | 0.41 ± 0.02 | 0.46 ± 0.1 | **0.54 ± 0.01** | 0.28 ± 0.01 |
| | DILO | **0.59 ± 0.03** | **0.97 ± 0.02** | 0.53 ± 0.02 | **0.64 ± 0.03** |
| Image 50 Demos | BCO | 0.00 ± 0.00 | 0.00 ± 0.00 | 0.00 ± 0.00 | 0.00 ± 0.00 |
| | SMODICE | 0.21 ± 0.02 | 0.40 ± 0.12 | 0.10 ± 0.04 | 0.02 ± 0.01 |
| | DILO | **0.76 ± 0.08** | **0.94 ± 0.02** | **0.25 ± 0.02** | **0.15 ± 0.01** |

Figure 2: Side-by-side comparison of LfO methods on state-only imitation vs image-only imitation. `DILO` shows noticeable improvement over existing LfO methods without hyperparameter tuning. Columns denote different suboptimal datasets.

## 5.3 Imitating from Human Trajectories for Robot Manipulation

**Setup**: Our setup utilizes a UR5e Robotic Arm on a tilted 1.93m × 0.76m Wind Chill air hockey table to hit a puck or manipulate tabletop objects. Puck detection utilizes an overhead camera, with additional environment details in Appendix 6.3. The set of tasks in this domain is designed to stress both 1) challenging inverse dynamics through complex striking motions and 2) partial state coverage

through the wide variety of possible paddle × puck positions and velocities. While baselines can struggle with compounding errors in one or both of these settings DILO's ability to side-step learning one-step models allow it to scale gracefully to these complexities.

**Tasks and Datasets**: We consider three tasks and 9 datasets for real-world experiments. Our tasks are:

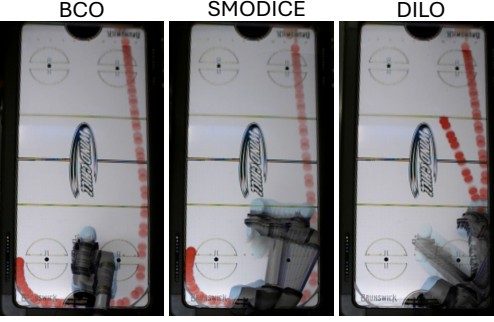

BCO · SMODICE · DILO

| Safe Object Manipulation | | | Puck-Striking | |
| --- | --- | --- | --- | --- |
| Few Trajectories | Fixed Start | Few Uniform | 20 expert | 10 expert |
| BCO 3/10 | 7/10 | **6/10** | 7/11 | 4/11 |
| SMODICE 2/10 | 1/10 | 0/10 | 5/11 | 4/11 |
| DILO **8/10** | **9/10** | 5/10 | **8/11** | **5/11** |
| Safe Object Manipulation (Cross-Embodiment) | | | Dynamic Puck Hitting | |
| Few Trajectories | Fixed Start | Few Uniform | 400 expert (touch) | 400 expert (hitting) |
| BCO 6/10 | 6/10 | **8/10** | 2/10 | 0/10 |
| SMODICE 1/10 | 1/10 | 1/10 | 6/10 | 4/10 |
| DILO **8/10** | **9/10** | **8/10** | **10/10** | **9/10** |

Figure 3: **Real Robot Experiments**: Table shows the (x/y) success rates as x successes in y trials for different methods on real-robot setup of air-hockey. For the dynamic puck-hitting task, we evaluate the number of touches made in addition to hitting behavior, which returns the puck in the opposite direction.

Figure 4: **Example of learned hitting behavior across algorithms**: Puck's (red) gradient shows movement across time for Dynamics Puck Hitting.

1) Safe Objective Manipulation: Navigate object safely to the goal without hitting obstacles. 2) Puck Striking: Hit a stationary puck 3) Dynamic Puck Hitting: A challenging task of hitting a dynamically moving puck. For the safe manipulation task, we investigate three datasets a) Few-Trajectories: 15 expert trajectory observations are given with uniform initial state b) Fixed-start-trajectories: 15 expert observation trajectories are provided to the agent with fixed start state. c) Few Uniform: 300 transitions are provided to the agent uniformly in state space. For Puck Striking tasks, we consider two observation datasets, one with 20 experts and the other with 10 experts. For Dynamic Puck hitting, we consider a dataset of 400 expert trajectories. The suboptimal datasets for all tasks contain the same amount of transitions as the expert dataset containing a mix of successes and failures. The datasets for all tasks are obtained by a teleoperation setup by humans, except for the cross-embodiment tasks where the humans demonstrate using their hands, and the state is detected using an overhead camera.

**Analysis:** Fig. 3 compares the success rate of Learning from Observation algorithms in settings with varying dynamics. Safe Object Manipulation presents a task with easy inverse dynamics modeling since the arm restricts its motion to move through the workspace. Consequently, BCO performs well when provided with good coverage of expert observations (few-uniform), but is still outperformed by DILO as a result of ignoring offline datasets to learn recovery behaviors. SMODICE shows poor performance consistently in tasks with small datasets—i.e. poor coverage. Puck striking presents both easy inverse dynamics and good state coverage, which may explain the comparable performance from BCO and SMODICE against DILO. On the other hand, Dynamic Puck Hitting is challenging both for inverse dynamics, because of the wide range of actions necessary to hit a moving puck, and for state coverage, where the range of possible paddle and puck positions is substantial. Fig. 4 demonstrates an example of learned puck hitting behavior. DILO handles both complexities gracefully, resulting in an impressive success rate over both baselines.

# 6 Conclusion

Offline Imitation from Observations provides a solution for fast adaptation of the agent to a variety of expert behaviors agnostic of the agent's action space. In this work, we propose a principled, computationally efficient, and empirically performant solution to this problem. Our work frames the problem as a particular distribution-matching objective capable of leveraging offline data. Using the principle of duality under a well-chosen but sufficient set of constraints, we derive an action-free objective whose training computational complexity is similar to an efficient offline RL algorithm. We show that the proposed method shows improved performance across a wide range of simulated and real datasets, learning from proprioceptive or image observations and cross-embodied expert demonstrations.

## Acknowledgements

We thank Siddhant Agarwal, Chang Shi, Carl Qi, Max Rudolph, Haoran Xu and Shuozhe Li for valuable comments on this work. This work has taken place in part in the Safe, Correct, and Aligned Learning and Robotics Lab (SCALAR) at The University of Massachusetts Amherst and Machine Intelligence and Decision Making Lab (MIDI) at The University of Texas. SCALAR research is supported in part by the NSF (IIS-2323384), AFOSR (FA9550-20-1-0077), the Center for AI Safety (CAIS), and the Long-Term Future Fund. HS and AZ are funded by NSF 2340651. The views and conclusions contained in this document are those of the authors and should not be interpreted as representing the official policies, either expressed or implied, of the Army Research Office or the U.S. Government. The U.S. Government is authorized to reproduce and distribute reprints for Government purposes notwithstanding any copyright notation herein.

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

# Appendix

## 6.1 Theory

### 6.1.1 Derivation for Action-free distribution matching

**Theorem 6.1.** *The dual problem to the primal occupancy matching objective (Equation 4) is given by the DILO objective in Equation 5. Moreover, as strong duality holds from Slater's conditions the primal and dual share the same optimal solution d\* for any offline transition distribution $\rho$ and any choice of mixture distribution ratio $\beta$.*

We start with the primal objective that matches distributions between the agent's visitation $d(s, s', a')$ and expert's visitation $d^E(s, s', a')$. As before $\rho$ denotes the visitation distribution of offline data.

$$\min_{\pi} \mathcal{D}_f(\text{Mix}_\beta(d^\pi(s, s', a'), \rho) \| \text{Mix}_\beta(d^E(s, s', a'), \rho)), \tag{7}$$

where for any two distributions $\mu_1$ and $\mu_2$, $\text{Mix}_\beta(\mu_1, \mu_2)$ denotes the mixture distribution with coefficient $\beta \in (0, 1]$ defined as $\text{Mix}_\beta(\mu_1, \mu_2) = \beta\mu_1 + (1 - \beta)\mu_2$.

Formulating the objective as a constrained objective in agent's visitation distribution $d$ allows us to create a primal objective that is a convex program. This is crucial in subsequently creating a dual objective that is unconstrained and easy to optimize.

$$\max_{d \geq 0} -\mathcal{D}_f(\text{Mix}_\beta(d, \rho) \| \text{Mix}_\beta(d^E, \rho))$$
$$\text{s.t} \sum_{a''} d(s', s'', a'') = (1 - \gamma)d_0(s', s'') + \gamma \sum_{s, a' \in \mathcal{S} \times \mathcal{A}} d(s, s', a')p(s''|s', a'), \ \forall s', s'' \in \mathcal{S} \times \mathcal{S}. \tag{8}$$

where the constraints above dictate the conditions that any valid visitation distribution $d(s', s'')$ needs to satisfy and are our proposed modifications to the commonly known *bellman flow constraints*.

Below we outline the derivation of how these specific constraints with the mixture distribution matching objective allows us to create a dual objective that is independent of expert's actions. Applying Lagrangian duality to the above constrained distribution matching objective, we can convert it to an unconstrained problem with dual variables $V(s, s')$ defined for all $s, s' \in \mathcal{S} \times \mathcal{S}$:

$$\max_{d \geq 0} \min_{V(s', s'')} -D_f(\text{Mix}_\beta(d, \rho)(s, s', a') \| \text{Mix}_\beta(d^E, \rho)(s, s', a'))$$

$$+ \sum_{s', s''} V(s', s'') \left( (1 - \gamma)d_0(s', s'') + \gamma \sum_{s, a'} d(s, s', a')p(s''|s', a') - \sum_a d(s', s'', a'') \right) \tag{9}$$

$$= \max_{d \geq 0} \min_{V(s, s')} (1 - \gamma)\mathbb{E}_{d_0(s, s')}\left[ V(s, s') \right] + \mathbb{E}_{s, s', a' \sim d}\left[ \gamma \sum_{s''} p(s''|s', a')V(s', s'') - V(s, s') \right]$$

$$- D_f(\text{Mix}_\beta(d, \rho)(s, s', a') \| \text{Mix}_\beta(d^E, \rho)(s, s', a')) \tag{10}$$

where the last equation uses a change of variable from $s', s''$ to $s, s'$ without loss of generality. Using a simple algebraic manipulation below, we can get rid of the inner maximization. We add and subtract the terms shown below:

$$= \max_{d \geq 0} \min_{V(s, s')} \beta(1 - \gamma)\mathbb{E}_{d_0(s, s')}\left[ V(s, s') \right]$$

$$+ \beta\mathbb{E}_{s, s', a' \sim d}\left[ \gamma \sum_{s'} p(s''|s', a')V(s', s'') - V(s, s') \right]$$

$$+ (1 - \beta)\mathbb{E}_{s, s', a' \sim \rho}\left[ \gamma \sum_{s''} p(s''|s', a')V(s', s'') - V(s, s') \right]$$

$$- (1 - \beta)\mathbb{E}_{s, a, g \sim \rho}\left[ \gamma \sum_{s'} p(s''|s', a')V(s', s'') - V(s, s') \right]$$

$$- D_f(\text{Mix}_\beta(d, \rho)(s, s', a') \| \text{Mix}_\beta(d^E, \rho)(s, s', a')) \tag{11}$$

As strong duality holds using Slater's conditions [52] (see [43] for a detailed account of strong duality in RL under visitation distributions). Using the fact that strong duality holds in this problem we can swap the inner max and min and rewrite an equivalent maximization under the mixture distribution:

$$
= \min_{V(s,s')} \max_{\texttt{Mix}_\beta(d,\rho)(s,s',a') \geqslant 0} \beta(1-\gamma)\mathbb{E}_{d_0(s,s')}\big[V(s,s')\big]
$$

$$
+ \beta\mathbb{E}_{s,s',a'\sim d}\left[\gamma\sum_{s'}p(s''|s',a')V(s',s'') - V(s,s')\right]
$$

$$
+ (1-\beta)\mathbb{E}_{s,s',a'\sim\rho}\left[\gamma\sum_{s''}p(s''|s',a')V(s',s'') - V(s,s')\right]
$$

$$
- (1-\beta)\mathbb{E}_{s,s',a'\sim\rho}\left[\gamma\sum_{s''}p(s''|s',a')V(s',s'') - V(s,s')\right]
$$

$$
- D_f(\texttt{Mix}_\beta(d,\rho)(s,s',a') \,\|\, \texttt{Mix}_\beta(d^E,\rho)(s,s',a')) \tag{12}
$$

In the following derivation, we will show that the inner maximization in Eq 12 has a closed form solution even when adhering to the non-negativity constraints. Let $y(s,s',a') = \mathbb{E}_{s''\sim p(s',a')}[V(s',s'')] - V(s,s')$.

$$
\max_{\texttt{Mix}_\beta(d,\rho)(s,s',a') \geqslant 0} \mathbb{E}_{s,s',a'\sim\texttt{Mix}_\beta(d,\rho)(s,s',a')}\left[\gamma\sum_{s''}p(s''|s',a')V(s',s'') - V(s,s')\right]
$$

$$
- D_f(\texttt{Mix}_\beta(d,\rho)(s,s',a') \,\|\, \texttt{Mix}_\beta(d^E,\rho)(s,s',a'))
$$

Now to solve this constrained optimization problem we create the Lagrangian dual and study the KKT (Karush–Kuhn–Tucker) conditions. Let $w(s,s',a') \triangleq \frac{\texttt{Mix}_\beta(d,\rho)(s,s',a')}{\texttt{Mix}_\beta(d^E,\rho)(s,s',a')}$, then the constraint $\texttt{Mix}_\beta(d,\rho)(s,s',a') \geqslant 0$ holds if and only if $w(s,s',a') \geqslant 0 \;\forall s,s',a'$.

$$
\max_{w(s,s',a')} \max_{\lambda \geqslant 0} \mathbb{E}_{s,s',a'\sim\texttt{Mix}_\beta(d^E,\rho)(s,s',a')}\big[w(s,s',a')y(s,s',a')\big] - \mathbb{E}_{\texttt{Mix}_\beta(d^E,\rho)(s,s',a')}\big[f(w(s,s',a'))\big]
$$

$$
+ \sum_{s,s',a'} \lambda(w(s,s',a') - 0) \tag{13}
$$

Since strong duality holds, we can use the KKT constraints to find the solutions $w^*(s,s',a')$ and $\lambda^*(s,s',a')$.

- **Primal feasibility**: $w^*(s,s',a') \geqslant 0 \;\forall\, s,a',a'$
- **Dual feasibility**: $\lambda^* \geqslant 0 \;\forall\, s,s',a'$
- **Stationarity**: $\texttt{Mix}_\beta(d^E,\rho)(s,s',a')(-f'(w^*(s,s',a')) + y(s,s',a') + \lambda^*(s,s',a')) = 0 \;\forall\, s,s',a'$
- **Complementary Slackness**: $(w^*(s,s',a') - 0)\lambda^*(s,s',a') = 0 \;\forall\, s,s',a'$

Using stationarity we have the following:

$$
f'(w^*(s,s',a')) = y(s,s',a') + \lambda^*(s,s',a') \;\forall\, s,s',a' \tag{14}
$$

Now using complementary slackness, only two cases are possible $w^*(s,s',a') \geqslant 0$ or $\lambda^*(s,s',a') \geqslant 0$.

Combining both cases we arrive at the following solution for this constrained optimization:

$$
w^*(s,s',a') = \max\left(0, f'^{-1}(y(s,s',a'))\right) \tag{15}
$$

Using the optimal closed-form solution $(w^*)$ for the inner optimization in Eq. (12) we obtain

$$\min_{V(s,s')} \beta(1-\gamma)\mathbb{E}_{d_0(s,s')}\big[V(s,s')\big]$$
$$+ \mathbb{E}_{s,s',a'\sim\mathtt{Mix}_\beta(d^E,\rho)(s,s',a')}\Big[\max\big(0,(f')^{-1}\big(y(s,s',a')\big)\big)\,y(s,s',a') - \alpha f\big(\max\big(0,(f')^{-1}\big(y(s,s',a')\big)\big)\big)\Big]$$
$$- (1-\beta)\mathbb{E}_{s,a\sim\rho}\Big[\gamma\sum_{s'} p(s'|s,a)V(s',s'') - V(s,s')\Big] \tag{16}$$

For deterministic dynamics, this reduces to the following simplified objective:

$$\min_{V(s,s')} \beta(1-\gamma)\mathbb{E}_{d_0(s,s')}\big[V(s,s')\big]$$
$$+ \mathbb{E}_{s,s',a'\sim\mathtt{Mix}_\beta(d^E,\rho)(s,s',a')}\Big[\max\big(0,(f')^{-1}\big(y(s,s',a')\big)\big)\,y(s,s',a') - f\big(\max\big(0,(f')^{-1}\big(y(s,s',a')\big)\big)\big)\Big]$$
$$- (1-\beta)\mathbb{E}_{s,a\sim\rho}\big[\gamma V(s',s'') - V(s,s')\big] \tag{17}$$

where $y(s,a,g) = \gamma V(s',s'') - V(s,s')$.

### 6.1.2 What does the utility function $V^*(s,s')$ represent?

Prior work [43] shows that for the regularized RL problem

$$\max_{d\geqslant 0} \mathbb{E}_{d(s,a)}[r(s,a)] - \alpha D_f(d(s,a) \,||\, d^O(s,a))$$
$$\text{s.t } \sum_{a\in\mathcal{A}} d(s,a) = (1-\gamma)d_0(s) + \gamma\sum_{(s',a')\in\mathcal{S}\times\mathcal{A}} d(s',a')p(s|s',a'), \ \forall s\in\mathcal{S}. \tag{18}$$

the dual optimizes for a Langrangian variable $V$ that represents a regularized optimal value function. This insight directly extends to our work with reward function set to zero, our Lagrangian variable learns only the regularized visitation probabilities under optimal policy.

It is easy to see why this is the case using the previous derivation. Following the derivation from the previous section, note that we had rewritten the inner maximization w.r.t the visitation distribution $d$, thus effectively getting rid of manipulating visitation distributions in the final objective. Our derivation above uses the following substitution shown in Eq 15 that holds as part of the closed form solution w.r.t inner maximization:

$$\frac{\mathtt{Mix}_\beta(d,\rho)(s,s',a')}{\mathtt{Mix}_\beta(d^E,\rho)(s,s',a')} = \max\Big(0,f'^{-1}(y(s,s',a'))\Big) \tag{19}$$

where $y = \gamma V(s',s'') - V(s,s')$. For deterministic dynamics, at convergence, the following holds for all $s,s',a'$ where $d^*(s,s',a') > 0$:

$$f'^{-1}(\gamma V^*(s',s'') - V^*(s,s')) = \frac{\mathtt{Mix}_\beta(d^*,\rho)(s,s',a')}{\mathtt{Mix}_\beta(d^E,\rho)(s,s',a')} \tag{20}$$

implying:

$$(\gamma V^*(s',s'') - V^*(s,s')) = f'\left(\frac{\mathtt{Mix}_\beta(d^*,\rho)(s,s',a')}{\mathtt{Mix}_\beta(d^E,\rho)(s,s',a')}\right) = -r_i(s,s',a') \tag{21}$$

The above relation makes the the interpretation of $V^*(s,s')$ clear. $(V^*(s,s') - \gamma V^*(s',s''))$ denotes the implied reward function $r_i(s,s',a')$ under which $V^*$ computes the maximum cumulative expected return, where $a'$ is the action that leads to $s''$. As shown above the the implied reward function $r_i(s,s',a') = -f'\left(\frac{\mathtt{Mix}_\beta(d^*,\rho)(s,s',a')}{\mathtt{Mix}_\beta(d^E,\rho)(s,s',a')}\right)$ is the divergence between expert stationary visitation distribution and agent's stationary visitation that is obtained after taking the action $a'$ from $s'$ and then acting optimally to match the expert visitation distribution. Note that the function $f'$ is non-decreasing as the function $f$ is convex from definition of $f$-divergences.

### 6.1.3 Analytical form of $f_p^*$ for $\chi^2$ divergence

For $\chi^2$ divergence, the generator function $f(x) = (x-1)^2$. $f'(x) = 2(x-1)$ and correspondingly $f'^{-1}(x) = \frac{x}{2} + 1$. Substituting $f'^{-1}(x)$ in definition of $f_p^*$:

$$f_p^*(x) = \max(0, f'^{-1}(x))(x) - f(\max(0, f'^{-1}(x))) \tag{22}$$

Since $x$ we substitute takes the form of residual $\text{residual} = \gamma\mathbb{E}_{s''\sim p(\cdot|s',a')}[V(s',s'')] - V(s,s'))$, the below pseudocode shows the implementation of $f_p^*$ for `DILO`.

```python
def f_star_p(self, residual, type='chi_square'):
    if type=='chi_square':
        omega_star = torch.max(residual / 2 + 1, torch.zeros_like(
    residual))
        return residual * omega_star - (omega_star - 1)**2
```

### 6.1.4 Intuitive understanding of DILO

To better understand this objective's behavior we consider the last two terms from Eq 5 in its expanded form below. We ignore the first term as it is simply pushing down $Q$-values at initial distribution of states, to prevent overestimation when learning from offline datasets.

$$\beta\mathbb{E}_{s,s',s''\sim\tilde{d}^E}\big[f_p^*(\gamma V(s',s'') - V(s,s'))\big] + (1-\beta) * \mathbb{E}_{s,s',s''\sim\rho}\big[f_p^*(\gamma V(s',s'') - V(s,s'))\big] \tag{23}$$
$$-(1-\beta)\mathbb{E}_{s,s',a'\sim\rho}\big[\gamma\mathbb{E}_{s''\sim p(\cdot|s',a')}\big[V(s',s'')\big] - V(s,s')\big],$$

Denote $r(s,s',a^E) = V(s,s') - \gamma V(s',s'')$ as the implicit expert reward of under a learned Q-function. The objective presents a clear intuition when we study the objective's behavior in different situations individually: (a) For samples from $\rho$, the objective pushes down the implicit reward to 0 as shown below:

$$\min_r \mathcal{L}(r) = \begin{cases} (1-\beta)\frac{r^2}{4}, \text{if } r < 2, \\ (1-\beta)r \text{ otherwise.} \end{cases} \tag{24}$$

(b) For samples from the expert distribution $\tilde{d}^E$, the objective ensures that reward is greater than equal to 2

$$\min_r \mathcal{L}(r) = \begin{cases} \beta(\frac{r^2}{4} - r), \text{if } r < 2, \\ 0 \text{ otherwise.} \end{cases} \tag{25}$$

It becomes clear now that `DILO` is implicitly learning a valid reward function that ensures higher discounted return for the expert compared to the suboptimal dataset by shaping $Q$-values directly.

### 6.2 Implementation

The algorithm for `DILO` can be found in Algorithm 1. We base the `DILO` implementation on the official implementation of pytorch-IQL that is based on IQL [44]. We keep the same network architecture as the original code and do not vary it across environments.

### 6.2.1 Imitation Learning with Proprioceptive Observations

Our experiment design is based on the benchmark from [13, 28] but we explain the setup here for completeness.

**Environments:** For the offline imitation learning experiments we focus on 9 locomotion and manipulation environments from the MuJoCo physics engine [18] comprising of Hopper, Walker2d, HalfCheetah, Ant, Kitchen, Pen, Door and Hammer to make a total of 24 datasets. The MuJoCo environments used in this work are licensed under CC BY 4.0 and the datasets used from D4RL are also licensed under Apache 2.0.

**Suboptimal Datasets:** We use the offline imitation learning benchmark from [28] that utilizes offline datasets consisting of environment interactions from the D4RL framework [19]. Specifically, suboptimal datasets are constructed following the composition protocol introduced in SMODICE [13]. The suboptimal datasets, denoted as 'random+expert', 'random+few-expert', 'medium+expert', and 'medium+few-expert' combine expert trajectories with low-quality trajectories obtained from the "random-v2" and "medium-v2" datasets, respectively. For locomotion tasks, the 'random/medium+expert' dataset contains a mixture of some number of expert trajectories ($\leqslant 200$) and $\approx 1$ million transitions from the "x" dataset. The 'x+few-expert' dataset is similar to 'x+expert,' but with only 30 expert trajectories included. For manipulation environments we consider only 30 expert trajectories mixed with the complete 'x' dataset of transitions obtained from D4RL. For the expert observation dataset we just 1 expert observation trajectory with length as the horizon of environment for our experiments. The detailed suboptimal dataset composition for different datasets can be found in table 2 below:

| Dataset | Data Points |
|---|---|
| All random+expert | 1m random transitions + 200 expert trajectories (horizon=1000) |
| All medium+expert | 1m medium transitions + 200 expert trajectories (horizon=1000) |
| All random+few-expert | 1m random transitions + 30 expert trajectories (horizon=1000) |
| All medium+few-expert | 1m medium transitions + 30 expert trajectories (horizon=1000) |
| Pen cloned+expert | 5e6 cloned transitions + 30 expert trajectories (horizon=100) |
| Pen human+expert | 5000 human transitions + 30 expert trajectories (horizon=100) |
| Door cloned + expert | 1m cloned transitions + 30 expert trajectories (horizon=100) |
| Door human + expert | 6729 human transitions + 30 expert trajectories (horizon=100) |
| Hammer cloned + expert | 1m cloned transitions + 30 expert trajectories (horizon=100) |
| Hammer human + expert | 11310 human transitions + 30 expert trajectories (horizon=100) |
| partial+expert | 136950 partial transitions + 30 expert trajectories (horizon=280) |
| mixed + expert | 136950 partial transitions + 30 expert trajectories (horizon=280) |

Table 2: Suboptimal dataset composition

**Expert Observation Dataset:** To enable imitation learning from observation, we use 1 expert observation trajectory obtained from the "expert-v2" dataset for each respective environment.

**Baselines:** To benchmark and analyze the performance of our proposed methods for offline imitation learning with suboptimal data, we consider different representative baselines in this work: BC [49], SMODICE [13], RCE [53], ORIL [48], IQLearn [50], ReCOIL [28]. SMODICE has been shown to be competitive [13] to DEMODICE [34] and hence we exclude it from comparison. SMODICE is an imitation learning method based on the dual framework, that optimizes an upper bound to the true imitation objective. ORIL adapts generative adversarial imitation learning (GAIL) [9] algorithm to the offline setting, employing an offline RL algorithm for policy optimization. The RCE baseline combines RCE, an online example-based RL method proposed by Eysenbach et al. [53]. RCE also uses a recursive discriminator to test the proximity of the policy visitations to successful examples. [53], with TD3-BC [54]. Both ORIL and RCE utilize a state-based discriminator similar to SMODICE, and TD3-BC serves as the offline RL algorithm. All the compared approaches only have access to the expert state-action trajectory.

The open-source implementations of the baselines SMODICE, RCE, and ORIL provided by the authors [13] are employed in our experiments. We use the hyperparameters provided by the authors, which are consistent with those used in the original SMODICE paper [13], for all the MuJoCo locomotion and manipulation environments.

In our set of environments, we keep the same hyper-parameters across tasks - locomotion, adroit manipulation, and kitchen manipulation. We train until convergence for all algorithms including baselines and we found the following timesteps to be sufficient for different set of environments: Kitchen: 1e6, Few-expert-locomotion: 500k, Locomotion: 300k, Manipulation: 500k

We keep a constant batch size of 1024 across all environments. For all tasks, we average mean returns over 10 evaluation trajectories and 7 random seeds. Full hyper-parameters we used for experiments

are given in Table 3. For policy update, using Value Weighted Regression, we use the temperature $\tau$ to be 3 for all environments.

Hyperparameters for our proposed off-policy imitation learning method `DILO` are shown in Table 3.

| Hyperparameter | Value |
| --- | --- |
| Policy learning rate | 3e-4 |
| Value learning rate | 3e-4 |
| $f$-divergence | $\chi^2$ |
| max-clip (Value clipping for policy learning) | 100 |
| MLP layers | (256,256) |
| $\beta$ (mixture ratio) | 0.5 |
| $\eta$ (orthogonal gradient descent) | 0.5 |
| $\tau$ (policy temperature) | 3 |

Table 3: Hyperparameters for `DILO` in imitation from proprioceptive observations.

### 6.2.2 LfO with Image Observations

We use robomimic [20] for our imitation with image observations experiments. The following two environments are used here (the description is taken from their paper and written here for conciseness):

**Lift**: Object observations (10-dim) consist of the absolute cube position and cube quaternion (7-dim), and the cube position relative to the robot end effector (3-dim). The cube pose is randomized at the start of each episode with a random z-rotation in a small square region at the center of the table.

**Can** Object observations (14-dim) consist of the absolute can position and quaternion (7-dim), and the can position and quaternion relative to the robot end effector (7-dim). The can pose is randomized at the start of each episode with a random z-rotation anywhere inside the left bin.

Robomimic provides three datasets and two modalities of observation (Proprioceptive, Images) for both environments above. The datasets are denoted by - MH (Multi-human), MG (Machine Generated), PH(Proficient-human). We use the MG and MH datasets as the suboptimal datasets in our task and PH as the source of expert observations. MH and MG datasets consists of 200 trajectories of usually suboptimal nature and we use 50 observation-only trajectory from PH datasets. This tasks is complex by the fact that expert-level actions are mostly unseen in the suboptimal dataset and the agent needs to learn the best actions that matches expert visitation from the suboptimal dataset. We implement all algorithms in the Robomimic codebase without any change in network architecture, data-preprocessing or learning hyperparameters. We tune algorithm specific hyperparameters in a course grid for BCO, SMODICE, and `DILO` to compare the best performance of methods independent of hyperparameters. For BCO, we tune the inverse dynamics model learning epochs between [1,5,10]. For SMODICE, we tuned discriminator learning epochs between [1,5], and gradient penalty between [1,5,10,20]. To control overestimation due to learning with offline datasets in `DILO` we consider a linear weighting $\lambda$ between the optimism and pessimism terms in Eq 5 inspired by prior work [28] as follows:

$$\min_{Q}(1-\lambda)\beta(1-\gamma)\mathbb{E}_{\tilde{d}_0}\big[V(s,s')\big] + \lambda\mathbb{E}_{s,s',a'\sim\text{Mix}_\beta(\tilde{d}^E,\rho)}\big[f_p^*(\gamma\mathbb{E}_{s''\sim p(\cdot|s',a')}\big[V(s',s'')\big] - V(s,s'))\big]$$

$$- \lambda(1-\beta)\mathbb{E}_{s,s',a'\sim\rho}\big[\gamma\mathbb{E}_{s''\sim p(\cdot|s',a')}\big[V(s',s'')\big] - V(s,s')\big],$$

The hyperparameters used for `DILO` can be found in Table 4. For the architecture specific hyperparameters we refer the readers to [20].

### 6.3 Robot Manipulation Experiments

Our setup for manipulation experiments is inspired by the robot air hockey environment [55] for applying `DILO` to physical robotics settings. Our setup utilizes a Universal Robotics 5 kilogram e-series (UR5e) 6-degree of freedom robotic arm on a fixed mount, a Robotiq parallel jaw gripper, a 1.93m × 0.76m Wind Chill air hockey table which is tilted at a 5.5 degree angle, and an overhead

| Hyperparameter | Value |
| --- | --- |
| max-clip (Value clipping for policy learning) | 100 |
| $\lambda$ (pessimism parameter) | 0.7 |
| $\beta$ (mixture ratio) | 0.5 |
| $\eta$ (orthogonal gradient descent) | 0.5 |
| $\tau$ (policy temperature) | 3 |

Table 4: Hyperparameters for DILO in imitation from image observations.

Sony Playstation Eye, a high framerate camera, which gathers $640 \times 480$ frames at 60 FPS, mounted to the ceiling to have a full view of the table. The paddle that is held by the robot end effector is 9.5cm in diameter and the puck is 6.3cm.

In this setup, the negative $x$ direction is oriented along the table, and the $y$ is across the table. The action space for the arm utilizes pose control in $x, y$ through an inverse kinematics controller accessible through the Universal

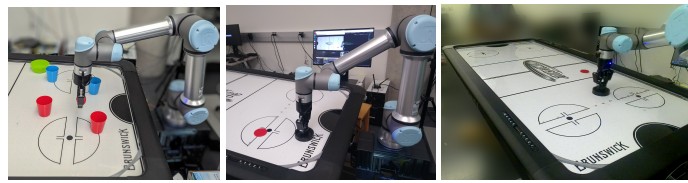

Figure 5: Tasks: Left: Place object and avoid obstacles. Center: Stationary Puck Striking. Right: Dynamic Puck Hitting

robotics real-time data exchange interface. Utilizing the serving command, the arm is controlled using delta positions clipped between 26cm in the x direction and 13cm in the right direction. These control limits are specified to prevent the robot from triggering force limits, which results in an emergency stop. Actions are taken at a 20Hz frequency to allow for rapid response to dynamic elements, such as hitting a falling puck.

The position and velocity of the end effector can be recovered through the real-time data exchange, but other objects like the puck or the hand require identification. This work utilizes an overhead camera running at 60Hz to locate these objects using a vision pipeline that relies on hue saturation value segmentation followed by object identification. This gives an $x_c, y_c$ coordinate in camera space, which we convert via OpenCV [56] homography to robot coordinates. This homography is computed by mapping the end effector positions given by the robot sensors to visual locations from the camera.

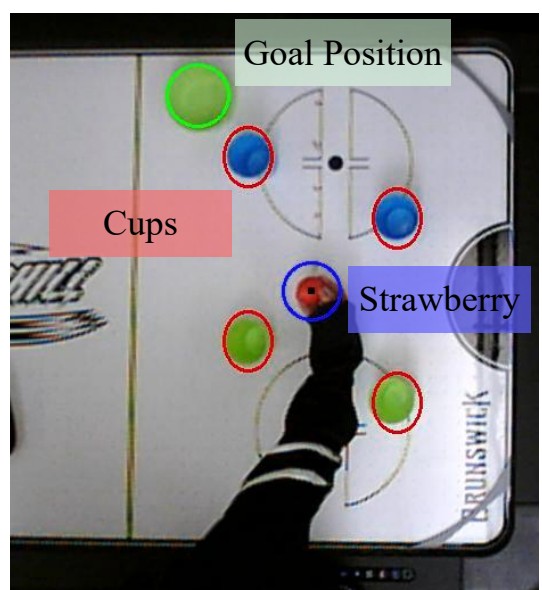

Figure 6: **Cross Embodiment Demonstration**: Tracking of the strawberry and obstacles for learning action-free.

We apply imitation learning from observations on several tasks built on top of the above robot setup. The following experiments are described here:

**Safe object manipulation**: This task involves moving a strawberry to a bowl while avoiding four cups placed in the workspace of the robot. The bowl is placed in the top right corner of the workspace, and the cups are placed in fixed locations. The success metric is the robot stopping above the bowl while making no contact with any of the cups. The test set involves 10 random starting locations for the end effector that are fixed between assessments. The observation space is the 2D end effector position, and the strawberry is initialized inside the gripper. Our suboptimal dataset

consists of 50 trajectories of 100 time steps on average where the robot is initialized in a random location, and the human moves the arm to a random different location, ignoring the positions of the cups or the bowl. In this setting, we investigated the following expert data, visualized in Figure 5:

- **Few Trajectories**: The expert data is drawn from a set where the expert is initialized in a random location, sometimes touching an obstacle, and must use the teleoperation system to avoid the obstacle and reach the goal. In this setting we used 15 expert trajectories.

- **Fixed start**: The expert is initialized at the opposite corner of the workspace, and navigates to the goal location following different paths using teleoperation. In this setting we used 15 trajectories.

- **Few Uniform**: Uses the same expert data as the Few Trajectories setting, but the dataset consists of randomly samples 300 transitions, where one trajectory is approximately 60 transitions of data.

- **Cross Embodiment Few/Fixed/Uniform**: The expert is a person holding the strawberry in his/her hand, visualized in Figure 6. They then move the strawberry tracked by the camera to the goal location while avoiding the obstacles, starting from random/fixed locations with 15 expert trajectories respectively or uniform with 300 transitions.

**Stationary Striking**: This task involves moving the end effector to strike a stationary puck. The success metric is the robot touching the puck. The test set involves 10 initializations of the puck position across the length of the table. The ensure uniformity across evaluations, the set of initialization locations of the puck are fixed across methods. The end effector is initialized at 0.38m from the base in the center of the table, so a success strike does not require backward motion. The observation space is the 2D end effector position and the tracked position of the puck. Our suboptimal dataset consists of 50 trajectories of 75 time steps on average where the robot is initialized at the start position, and the human moves the arm in a random, vaguely striking pattern.

In this setting we used an expert dataset of 400 trajectories where the expert uses mouse teleoperation to strike the puck. The expert efficiently strikes the puck in a single motion. We visualize the expert striking and the puck position in Figure 5. We show the learned action vectors for all algorithms and tasks fixed start (Figure 9), Few Uniform (Figure 8) and Few trajectories (Figure 10).

**Dynamic Hitting**: This task involves hitting a puck dropped from the top of the table. Because the table is set at an angle, this will cause the puck to fall with increasing acceleration towards the opposite side. The setup is visualized in Figure 5. The test set involves 10 initializations of the puck position dropped from positions across the length of the top of the table. The locations of the 10 puck drops are fixed using indicators across methods to give fair evaluation, and the arm is initialized in the center of the table, 0.68m from the base. The observation space is the 2D end effector position and 2D end effector velocity and the history of the last 5 tracked positions of the puck relative to the position of the end effector. Our suboptimal dataset consists of 50 trajectories of 200 time steps on average where the robot is initialized at the start position, and the human moves the arm around the puck without striking it.

We utilize two success metrics for this task: 1) touching: a trajectory is considered successful if the agent touches the puck. 2) hitting: the puck must have velocity in the opposite direction that it was dropped. This task is especially challenging for existing methods because of the long sequence of actions necessary to position the paddle properly, and the high level of both precision and timing: even a few millimeters of error or a movement at the wrong time will result in a failure, especially for hitting. Previous work has observed that this task is challenging even for humans, who often require several tries of practice, and many dataset trajectories consist of many inaccurate hits. In this domain, Behavior cloning only achieves 30% success at touching the puck, and Implicit Q-learning, a popular offline RL method, can only achieve 60% success, even though it employs a hand-designed reward function.

We used the implementation details from the proprioception task with the difference that in all the real-robot tasks we tune the following parameters across different methods: For BCO, we tune the

inverse dynamics model learning epochs between [1,5,10]. For SMODICE, we tuned discriminator learning epochs between [1,5], and gradient penalty between [1,5,10,20]. For DILO we tune the conservatism parameter from previous section between [0.5,0.6,0.7,0.8].

In this domain, these challenges appear to be empirically validated in the performance of the baseline methods. We hypothesize that the accumulation of error over long horizons in other learning from observation methods results in poor performance, as visualized in Table 3 and Figure 4. For methods like BCO, learning the necessary inverse dynamics to interpolate the long sequence of actions for a successful strike is impractical, resulting in behavior that appears listless. On the other hand, while the SMODICE discriminator rewards are able to occasionally match the visitation distribution of the expert, there is an exponential explosion of possible combinations of puck history and paddle positions, resulting in poor generalization: on the left half of the table, SMODICE is unable to hit the puck.

The hitting scenario involves two settings, both using mouse teleoperation to control the puck: one where the human strikes the puck only once and then the trajectory ends, which utilizes a dataset of 50 trajectories, and an expert dataset of 850 trajectories where the human keeps hitting the puck repeatedly for up to 2000 timesteps. The task is challenging for human, so trajectories average only 500 timesteps and cleaned so that human mistakes are removed from the expert dataset. We visualize the expert hitting and the puck position in Figure 7.

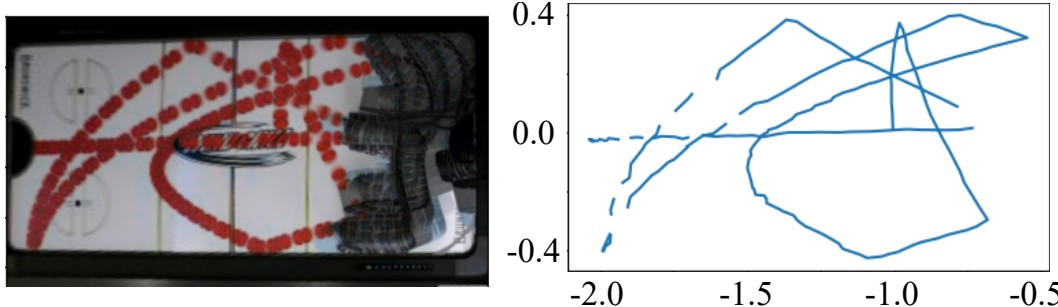

Figure 7: **Expert Hitting**: Visualization of one trajectory of puck tracking and hitting by the expert. **Right**: stacked frames of the environment. **Left**: puck position in robot coordinates

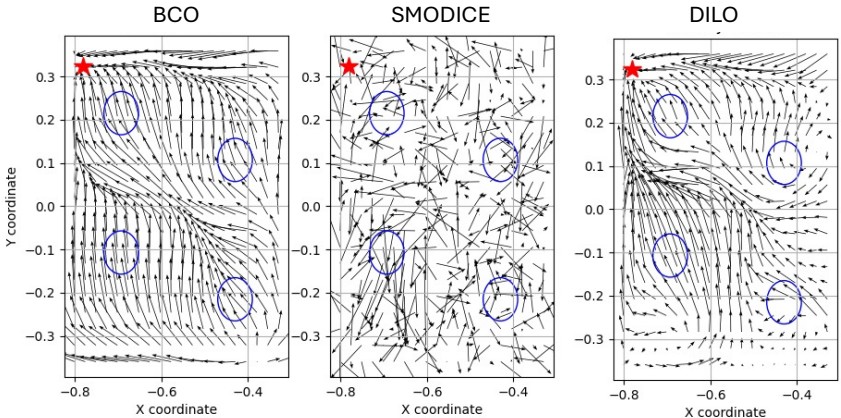

Figure 8: Action Vectors qualitatively showing the next x-y action for the safe manipulation with uniform sampled transitions. BCO generalizes incorrectly at a number of locations producing policies that hit obstacles. DILO learns to mimic expert's intent better demonstrating signs that it has learned to avoid obstacles by the arrows around

## 6.4 Limitations

Learning from Observation is a challenging setting, and while DILO makes some key assumptions in order to achieve good performance. First, matching distributions becomes exponentially more challenging in the dimensionality of the state space. In this work, while DILO outperforms baselines

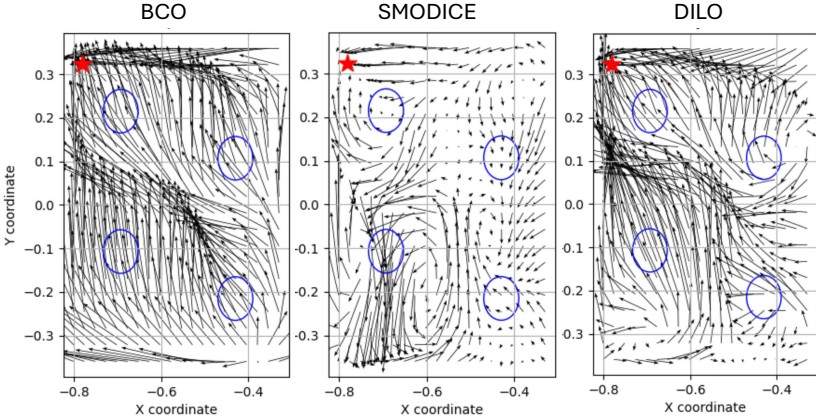

Figure 9: Action Vectors qualitatively showing the next x-y action for the safe manipulation with fixed start state. BCO generalizes incorrectly at a number of locations producing policies that hit obstacles. DILO learns to mimic expert's intent better demonstrating signs that it has learned to avoid obstacles by the arrows around

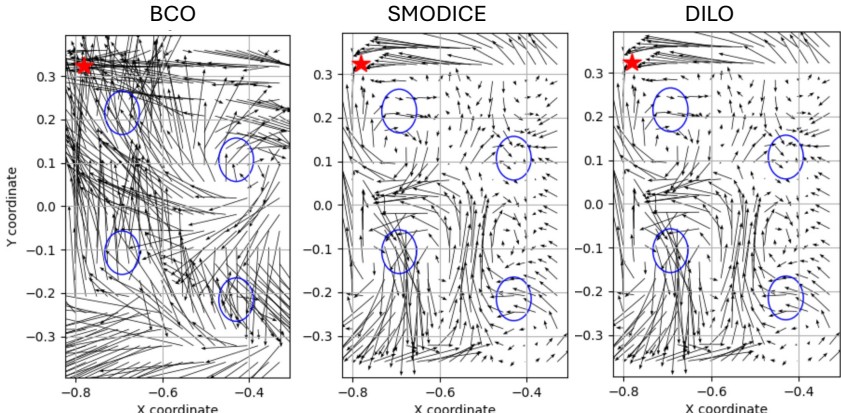

Figure 10: Action Vectors qualitatively showing the next x-y action for the safe manipulation with few trajectories. BCO generalizes incorrectly at a number of locations producing policies that hit obstacles. DILO learns to mimic expert's intent better demonstrating signs that it has learned to avoid obstacles by the arrows around

in the Expert Image observations, it still shows limited performance. Second, while learning from observations opens the door for good performance without expert actions, the expert observation space must match that of the agent. In some video settings, this is not the case, ex. the agent might use a fixed camera when the human is egocentric, or vice versa. Finally, DILO utilizes the conservatism parameter $\tau$ to regulate the degree of extrapolation from the algorithm. In some settings, the values can diverge, resulting in $V^*$ taking on values that might be too large to be used for learning the downstream policy. Adaptively selecting $\tau$ to maximize extrapolation while avoiding divergence is an area of active investigation.

### 6.4.1 Failure Cases

While DILO outperforms other methods in overall success rate, the failure modes can differ. In general, DILO tends to be conservative in what actions it takes, learning motions that might be slower than BCO or may get stuck before arriving at the goal. In low-dim observation settings, DILO can also exhibit "dead zone" behavior, where the model becomes mostly unresponsive. Below we detail some of the exact error modes in particular tasks:

**Safe Manipulation**: While DILO and BCO have comparable success rates, the two algorithms fail in different ways. BCO tends to take large actions while ignoring obstacles to reach the goal, while DILO takes more conservative actions. Thus, while BCO might fail by knocking over a cup, DILO will tend to fail to reach the goal. Because this is a low data setting, both algorithms BCO and DILO

can end up coming close to the cups or brushing them gently. As a side note, SMODICE fails at even reaching the goal in most cases in this task, possibly because of this low data setting.

**Striking**: This domain is challenging because of the narrow data regime, and all methods tend to struggle in similar ways. The most common consequence of low data arises through sensitivity to the $x$ location of the puck (along the table). While intuitively, striking behavior should be relatively invariant for a fixed $y$ (horizontal position on the table), slight variation in $x$ from the dataset can result in a policy that moves the arm in the opposite direction of the puck, probably due to errors in extrapolation. In addition, striking is a dynamic behavior that requires a precise combination of forward and horizontal actions. Even a slight error in the ratio can result in a near miss. Finally, `DILO` tends to learn more conservative policies and, in some locations, may not not strike the puck with much force. However, because of the low data coverage, this issue is endemic to all the learned policies.

**Hitting**: The primary challenge of achieving a hit in this task is the precise alignment of the paddle to the puck. While `DILO` performs well, it is not perfectly accurate, resulting in touches that bounce off the side of the paddle. This challenge is endemic to all policies. Additionally, the conservatism of `DILO` actions appear when it moves under the puck, where it tends to move slowly, and dropping the puck too quickly can result in `DILO` failing to reach the puck. As a result, while `DILO` is likely to succeed at the first hit, it can struggle to generate multiple hits because this can require rapid side-to-side movement. These issues are largely endemic to all the learned policies, where SMODICE tends to be even less precise, and BCO struggles to learn to strike, though it can occasionally position under the puck.

Visualizations of the failure modes can be seen in the project website.

## 6.5 Limitations of DILO

Our proposed method is limited by the assumption of matching visitation distributions *in the observation space of the agent and expert* rather than a meaningful semantic space, but we hope that with improvement in universal representations, this limitation is lifted by distribution matching in compact representation space. Our work assumes that expert's optimality, but in reality, experts demonstrate a wide range of biases. We leave this extension to future work. Finally, we demonstrate the failure modes of our method and further limitations in Appendix 6.4.

