# OpenReview forum: "A Dual Approach to Imitation Learning from Observations with Offline Datasets"
_robot-learning.org/CoRL/2024/Conference — CoRL 2024_

### Official Review · Reviewer_2kPL · 2024-07-19
**Review of Submission 200**

**Originality:** 4
**Technical Quality:** 4
**Clarity Of Presentation:** 4
**Potential Impact:** 4
**Recommendation:** 4
**Confidence:** 3

**Review:**

Overall, this paper contains a solid contribution to the field of imitation learning.


Strengths
+ This paper focuses on one of the most challenging settings in imitation learning, offline imitation learning from observations, where the agent has access to an offline dataset of its own trajectories of arbitrary quality and a few task-relevant expert trajectories only containing observations. Given this focus, the results are very impressive.
+ The paper is well-written. There are ample citations justifying the approach and its design decisions, detail regarding the formulation of DILO, techniques to improve training practicality, and intuitive explanations.
+ The evaluation is very extensive and displays the success of the proposed technique. It is clear that the authors were thorough in their evaluation by including results in typical Mujoco-like benchmarks, LfO from images, and finally from real-world human demos.



Weaknesses
- Table 1 displays an abundance of results and DILO is outperforming baselines on average. However, as SMODICE is a close competitor, it would be beneficial to add additional description on why SMODICE is able to outperform DILO in some instances (e.g., random + expert) and not in others (e.g., cloned + expert). Also, could the authors comment on why the deviations between DILO and SMODICE are very large in domains like pen, door, and hammer?

**Quality Of The Limitations Section:**

3

**Questions For Rebuttal:**

- Could you please address the weakness noted above?
-  DILO achieves similar results for Safe Object Manipulation across the teleop and cross-embodiment demonstration. I would expect it would be easier to learn from the teleop demonstrations. Could you comment on why the performance is similar and if this was expected/designed for?

**Robotics Focus:**

4

**Summary Of Paper:**

This paper presents a novel approach for learning from observation (LfO) without access to a simulator. The authors start by motivating their approach, noting that many prior works fail in the offline LfO space. The authors then present their approach (Dual Imitation Learning from Observations or DILO), which directly learns a utility function that quantifies how state transitions impact the agent's divergence from the expert's visitation distribution. Importantly, the method is backed with several theoretical results, derivation, and ample citations, and explained both intuitively and in-depth. The authors conclude by displaying an abundance of results, both in synthetic domains and real-world, displaying that DILO presents a novel high-performing algorithm for learning robot behavior.

**Summary Of Recommendation:**

The paper presents a clear advancement for offline learning from observation, contains a strong mathematical basis behind the algorithm, and has ample results across synthetic and real-world domains.

---

### Official Review · Reviewer_5eEJ · 2024-07-21
**Initial review**

**Originality:** 3
**Technical Quality:** 2
**Clarity Of Presentation:** 3
**Potential Impact:** 2
**Recommendation:** 3
**Confidence:** 3

**Review:**

Strengths:

The problem of learning from action-free videos is both highly relevant and significant to the field of robotics research. The proposed method addresses this problem in a novel and reasonable manner.

The paper is well-structured and thoroughly written, providing sufficient technical details to clearly convey the overall idea and methodology. The inclusion of mathematical formulas and proofs adds rigor and credibility to the proposed approach.

The method is validated through real-world hardware experiments, demonstrating practical applicability.

Weaknesses:

The effectiveness of the proposed approach is contingent on a substantial amount of training data that adequately covers the state transition space, which may be a limitation in terms of data requirements.

The performance of the proposed method appears to be less effective compared to simpler behavior cloning approaches, potentially indicating room for improvement.

The real-world implementation relies on object state representations that depend on robust detection algorithms. This dependency could pose challenges when applying the method to more complex tasks or environments.

**Quality Of The Limitations Section:**

2

**Questions For Rebuttal:**

Would the proposed method be sensitive to the quality of demonstrations? It would be valuable to analyze how policies learned from demonstrations of varying quality (e.g., using datasets from Robomimic[1]) perform on the tasks.

The performance for Robomimic benchmark seems to be not ideal, could authors give details analysis and explanations for this?

Besides the proposed tasks, demonstrating the effectiveness of the approach on other benchmarks (e.g., RLBench) would further strengthen the paper.

[1] What Matters in Learning from Offline Human Demonstrations for Robot Manipulation, CoRL’21.

[2] RLBench: Robot Learning Benchmark, RA-L'20.

**Robotics Focus:**

4

**Summary Of Paper:**

The paper works on the problem of learning robot manipulation skills from actionless expert demonstration videos. The proposed method first learns a utility function over the state transitions from the offline dataset, and then extracts the policy through maximizing the expected returns of the learned utility function. Experiments are performed in simulation and real world.

**Summary Of Recommendation:**

My rating is based on insufficient evaluation and weak performance

---

### Official Review · Reviewer_U3CE · 2024-07-27
**The paper proposes a novel learning-from-observations method with substantial technical contributions. However, presentation quality needs to be improved.**

**Originality:** 4
**Technical Quality:** 4
**Clarity Of Presentation:** 2
**Potential Impact:** 3
**Recommendation:** 3
**Confidence:** 3

**Review:**

**Strengths**:

* Learning from observations and suboptimal action-labelled data is an interesting and important problem, which is well motivated in the paper.
* Learning a multi-step utility function instead of an inverse dynamics model or a discriminator is a nice idea and is motivated well throughout the paper.
* An action-free objective that allows learning the utility of transitioning between states based on the visitation distribution of an expert is a valuable contribution.
* Derivations provided in the appendix are useful for understanding the method.
* Experiments in various settings effectively demonstrate the advantages of DILO compared to alternative methods for learning from observations.

**Weaknesses**:

* The paper is quite convoluted with many ideas repeated multiple times without a clear and concise overview, making it hard to follow.
* The paper is poorly written overall, with numerous grammatical errors and incomplete sentences.
* It is not discussed what "suboptimal" or "arbitrary" data actually means. How different can the distribution of this suboptimal data be from the expert data distribution?
* The experimental setting is not well described. It is not clear what different Suboptimal Datasets in Table 1 represent and how big they are. Adding exact numbers of action-free and action-labeled data points in the first column of Table 1 would really help.
* A more detailed discussion about the results presented in Table 1 would be helpful in understanding settings where DILO really shines (e.g. dimensionality of state representations, dataset size and quality).
* A more detailed discussion about using image observations and how the performance could be increased would be helpful.

**Quality Of The Limitations Section:**

2

**Questions For Rebuttal:**

* How different can the distribution of suboptimal data be from the expert data distribution while still being able to extract a policy from the learned utility function?
* How well does DILO scale with increasing dataset size?
* Could DILO be extended to a multi-task (or goal-conditioned) setting?
* How can the performance of DILO with image observations be improved? What properties should a representation learned from visual inputs have to be easily applicable to DILO?
* What theoretical properties of DILO you are referring to on lines 283-284?

**Robotics Focus:**

4

**Summary Of Paper:**

The paper proposes a novel method for learning policies from expert observation trajectories without action information and from an action-labeled dataset of arbitrary quality. An action-free distribution matching objective is used to learn a utility function that quantifies the utility of transitioning between states based on the visitation distribution of an expert. The policy is extracted from the learned utility function using value-weighted regression on the action-labeled dataset. Experiments show improved performance over alternative methods that address the problem of learning from observations in a variety of settings.

**Summary Of Recommendation:**

After rebuttals, I would recommend accepting this paper for publication.

---

### Author Rebuttal · Authors · 2024-08-05

We appreciate the reviewers for their insightful comments. We are encouraged to see that all reviewers found our problem **well motivated** and our algorithm **novel and theoretically sound**, particularly reviewer 2kPL finding our work to be **an important addition to the field of imitation learning**. We are motivated that reviewer U3CE and 2kPL found our **simulated and real world experiments abundant and convincing**. We summarize the issues raised by reviewers in this comment and provide a summary of response here. We also attach the updated paper with changes marked in red in response to reviewers concerns. A detailed response can be found in individual reviewer response:

1. Reviewer U3CE raised the concern of grammatical errors and clarity of presentation

We have worked on this concern and uploaded a revised version with improved writing and corrected grammatical errors. We have also added complete experimental details in the appendix. We welcome further comments from the reviewer if they have any particular things in mind that were not addressed.

2. Reviewer 5eEJ raised the following concerns:
* *The algorithm requires coverage of the state transition states and lots of data.*

	We detail in the response that **this is not the case both theoretically and experimentally**, as our experiments consider datasets in the order of size 1k to 1 million transitions.

* *The algorithm’s performance is less effective than behavior cloning.*

	We detail in the response that **this is false**. Our algorithm consistently outperforms behavior cloning in Table 1 and Fig. 2.

* *Demonstrating effectiveness on more benchmark like RLBench can further strengthen paper.*

	We consider **16 environments with 42 diverse datasets, substantially more than prior works in the domain**. RLBench does not readily provide suboptimal datasets suitable for our setup and requires resources and time beyond the rebuttal period.

* *Real-world implementation required good state-representation.*

	**Our experiments show that DILO can learn from observation streams like images (Section 5.2)** and we argue that state representations are successfully used in many domains today (autonomous driving) and have progressed considerably in the last few years (e.g. developments in segment anything).

---

### Decision · Program_Chairs · 2024-09-04

**Decision:**

Accept

**Comment:**

**Before Rebuttal**

Strengths.
This paper studies learning from observations with action-free videos, an important problem to solve given the abundance of such data. The proposed method is novel, with strong theoretical motivations. Experimental results in both simulation and reality show the benefit of the method over alternative approaches. As well as comparing to alternative methods for learning from observation without action data, comparisons to methods which additionally have access to action data offer interesting insights.

Weaknesses.
In some places, the paper is difficult to follow and understand due to low quality of the writing. There is insufficient information in the method and experiments to appreciate the intended contributions, and to understand what the key conclusions are from the results. The real-world experiments assume that low-dimensional state representations are available, which are challenging to obtain in practice, and so there is no evidence that the method performs well in the real world directly from visual observations.

---

**After Rebuttal**

Following the reviews, authors provided a revised paper which has now improved in clarity and addressed some of the concerns raised by the reviewers. Although the real-world experiments are limited due to the need for low-dimensional states, overall this is only a minor issue given the strong theoretical contribution of the paper. The AC and reviewers discussed the paper together, after which there is now unanimous agreement amongst reviewers that the paper should be accepted.